# Straightforward Access to a New Class of Dual DYRK1A/CLK1 Inhibitors Possessing a Simple Dihydroquinoline Core

**DOI:** 10.3390/molecules28010036

**Published:** 2022-12-21

**Authors:** Mihaela-Liliana Ţînţaş, Ludovic Peauger, Florent Alix, Cyril Papamicaël, Thierry Besson, Jana Sopková-de Oliveira Santos, Vincent Gembus, Vincent Levacher

**Affiliations:** 1INSA Rouen Normandie, Univ. Rouen Normandie, CNRS UMR 6014 COBRA, FR 3038, F-76000 Rouen, France; 2VFP Therapies, 15 rue François Couperin, 76000 Rouen, France; 3UNICAEN, CERMN (Centre d’Etudes et de Recherche sur le Médicament de Normandie), Normandie Univ., Bd Becquerel, F-14032 Caen, France

**Keywords:** Alzheimer’s disease, Down syndrome, *h*DYRK1A, *h*CLK1, dihydroquinoline, dual inhibitor, antioxidant, radical scavenger, molecular docking

## Abstract

The DYRK (Dual-specificity tyrosine phosphorylation-regulated kinase) family of protein kinases is involved in the pathogenesis of several neurodegenerative diseases. Among them, the DYRK1A protein kinase is thought to be implicated in Alzheimer’s disease (AD) and Down syndrome, and as such, has emerged as an appealing therapeutic target. DYRKs are a subset of the CMGC (CDK, MAPKK, GSK3 and CLK) group of kinases. Within this group of kinases, the CDC2-like kinases (CLKs), such as CLK1, are closely related to DYRKs and have also sparked great interest as potential therapeutic targets for AD. Based on inhibitors previously described in the literature (namely TG003 and INDY), we report in this work a new class of dihydroquinolines exhibiting inhibitory activities in the nanomolar range on *h*DYRK1A and *h*CLK1. Moreover, there is overwhelming evidence that oxidative stress plays an important role in AD. Pleasingly, the most potent dual kinase inhibitor **1p** exhibited antioxidant and radical scavenging properties. Finally, drug-likeness and molecular docking studies of this new class of DYRK1A/CLK1 inhibitors are also discussed in this article.

## 1. Introduction

Alzheimer’ s disease (AD) is a neurodegenerative disease, clinically characterized by a progressive deterioration of cognitive functions, leading to an inexorable decline in functional abilities. In addition, AD patients often present neuropsychiatric and behavioral symptoms such as depression, restlessness and hallucinations leading to a deterioration of their quality of life [1]. The pathogenesis of AD is multifactorial, characterized by specific brain anatomical and pathological abnormalities: a significant extent of oxidative stress associated with the presence of extracellular deposits of amyloid-β(Aβ) peptide and intraneuronal neurofibrillary tangles [2,3,4].

Dual-specificity tyrosine phosphorylation-regulated kinase 1A (DYRK1A) is a protein kinase that is abnormally expressed in many diseases such as Down syndrome or AD [5,6,7]. In the literature, it was reported that DYRK1A is able to phosphorylate both APP on Thr668 [8] and presenilin-1 on Thr354 [9]. When the latter is hyper-phosphorylated, the activity of γ-secretase, one of the proteases that cleave APP, is increased. This leads to an abnormally high production of β-amyloid peptides, the main components of amyloid plaques, that are responsible for neurodegeneration [10,11].

DYRK1A was also found to be involved in the hyperphosphorylation of Tau protein (Tubulin-Associated Unit) [12,13,14,15,16], which triggers Tau to dissociate from neuronal microtubules and, shortly afterwards, causes microtubules to disassemble while leading to the formation of neurofibrillary tangles and final neurodegeneration.

Furthermore, the CDK-like kinase (CLK) family, composed of four isoforms: CLK1, CLK2, CLK3 and CLK4, is also an interesting target for treating AD [7,17,18,19,20]. As for DYRK1A, they are dual-specificity kinases. In particular, CLK1 is likely involved in AD by phosphorylating the serine residue in serine/arginine-rich (SR) proteins [19]. SR proteins are a family of splicing factors involved in the alternative splicing of Tau.

Although some natural compounds, such as epigallocatechin-3-gallate (EGCG) [21], harmine [22], and leucettamine B [23], have been identified as potent inhibitors of DYRK1A or CLK1, significant research effort has been devoted to developing synthetic DYRK1A and/or CLK1 inhibitors based on relatively sophisticated heterocyclic scaffolds [7,19,24,25,26,27,28]. A large majority of these kinase inhibitors are type I, that is, they bind to the active form of the kinase in the ATP pocket [29]. 

As part of a research program seeking new and readily accessible classes of DYRK1A and/or CLK1 kinase inhibitors, our attention was drawn to INDY [(*Z*)-1-(3-Ethyl-5-hydroxy-2(3*H*)-benzothiazolylidene)-2-propanone)] and its derivatives such as TG003 (its synthetic precursor) and ProINDY (its prodrug), reported by Ogawa [30] and Hagiwara [31], respectively (Figure 1). These DYRK1A/CLK1 inhibitors are all derived from 5-methoxy-2-methylbenzothiazole and show significant structural analogy to the 3-acetyl 7-substituted 1,4-dihydroquinoline derivatives. It is worth mentioning that this new class of dual DYRK1A and CLK1 inhibitors based on a biooxidizable 1,4-dihydroquinoline scaffold may also undergo oxidation into quinolinium salts. That could lead to antioxidant properties going hand in hand with the attenuation of oxidative stress in brain tissues. In the present article, we report the synthesis of these biooxidizable 1,4-dihydroquinoline derivatives, a preliminary in vitro biological evaluation toward DYRK1A and CLK1 kinases, and an examination of their redox properties in various biological media.

## 2. Results and Discussion

### 2.1. Synthesis of the Studied Dual hDYRK1A and hCLK1 Inhibitors

Apart from commercial quinoline **6a**, 3-acetyl quinoline derivatives **6b**–**h** were obtained by reacting the corresponding *o*-nitrobenzaldehyde derivatives **3b**–**h** with 4,4-dimethoxybutan-2-one **4** in the presence of tin chloride as a reducing reagent (Figure 1). This domino nitro reduction-Friedländer cyclisation, previously reported in the literature for the preparation of quinoline-3-carboxylic acid esters [32], provided access to the required 3-acetyl quinoline derivatives **6b**–**h** in a one-step procedure from readily available and stable starting materials.

We then focused on the preparation of compounds **6i**–**m** to investigate the influence of the electron withdrawing group (EWG) at C-3 position of the quinoline ring on the inhibitory activity of the DYRK1A and CLK1 kinases. To this end, the 3-cyano-7-methoxy quinoline **6i** previously reported in our group [33] was reacted with chlorotrimethylsilane and anhydrous methanol [34] to provide methyl ester **6j**. Quinoline ester **6k** was obtained by reacting thionyl chloride on the reported quinoline 3-carboxylic acid **5** [35] in ethanol. Finally **6l**,**m** were prepared from quinoline derivative **5** using T3P^®^ and CDI amide coupling reagents, respectively. Subsequently, quinoline derivatives **6a**–**m** were subjected to a classical quaternization reaction with alkyl halides to afford the desired quinolinium salts **2a**–**p**, which were then successfully regioselectively reduced with BNAH to the corresponding 1,4-dihydroquinolines **1a**–**p**. With the target compounds **1a**–**p** in hand, readily obtained in a few steps, and in most cases, in a good overall yield, we embarked on an initial biological evaluation of these dihydroquinoline scaffolds as potential new DYRK1A and CLK1 inhibitors (Table 1).

### 2.2. In Vitro Evaluation on hDYRK1A and hCLK1 Kinases

We initially focused on the influence of the substitution patterns at the benzene ring by comparing the *h*DYRK1A and *h*CLK1 inhibitory activities of a set of dihydroquinolines **1a**–**h** differently substituted at C-6 and C-7 (EWG = COMe and R^3^ = Me). To our delight, compound **1h** displayed interesting three-digit nanomolar *h*DYRK1A and *h*CLK1 inhibitory activities (Entry 8) comparable to those of INDY and TG003, whereas compounds **1a**–**d** showed no inhibition on either of the two kinases (Entries 1−4). This first series revealed that a methoxy group at C-7 is essential while an additional substituent at C-6 appears to be deleterious, resulting in a complete loss of the inhibitory activity (Entries 6,7). The influence of the EWG at C-3 position, which is crucial to ensure good stability of the 1,4-dihydroquinoline, was then assessed by replacing the acetyl group in **1h** by a variety of EWGs (nitrile, amides, esters). The resulting compounds **1i**–**m** turned out to be inactive toward both *h*DYRK1A and *h*CLK1 kinases (Entries 9−13), with the sole exception of the ester derivatives **1j**,**k** that exhibited micromolar *h*CLK1 inhibitory activities (Entries 10,11). After demonstrating that both an acetyl group at C-3 position and a methoxy group at C-7 position are required, we turned our attention to the role of the alkyl group on the nitrogen dihydroquinoline scaffold. Compared to *N*-methyl dihydroquinoline **1h**, *N*-benzyl and *N*-phenethyl dihydroquinolines **1n,o** exhibited lower *h*DYRK1A inhibitory activities, while displaying comparable three-digit nanomolar *h*CLK1 inhibitory activities (Entries 14,15). Gratifyingly, *N*-propyl dihydroquinoline **1p** turned out to be 2.5-fold more potent than *N*-methyl dihydroquinoline **1h** toward *h*DYRK1A and 8-fold more potent than *N*-methyl dihydroquinoline **1h** toward *h*CLK1 (Entries 8,16). To complete this SAR analysis, we sought to determine whether or not the enamine function is a crucial element for this new class of dual DYRK1A/CLK1 inhibitors. To tackle this issue, quinolinium salts **2h** and quinoline **6h** were also evaluated for inhibition of *h*DYRK1A/*h*CLK1. None of these synthetic intermediates showed inhibitory activities on *h*DYRK1A/*h*CLK1 kinases (IC_50_ > 10 mM). These observations led us to conclude that, in addition to the acetyl group at C-3 and the methoxy group at C-7, the enamine-like nitrogen atom is also a prerequisite for inhibition of *h*DYRK1A/*h*CLK1.

### 2.3. Evaluation of the Antioxidant and Radical Scavenging Activities of 1p

Oxidative stress is also an important hallmark of AD [36,37,38,39,40] and is closely linked with the formation of both the neurofibrillary tangles and the amyloid plaques.

A further interesting aspect of this novel class of *h*DYRK1A/*h*CLK1 inhibitors is related to the potential of 1,4-dihydroquinolines **1** to prevent oxidative stress by exhibiting in vivo antioxidant and radical scavenging properties during their conversion to the corresponding quinolinium salts **2** in the Central Nervous System (CNS) [35,41].

To evaluate the ability of 1,4-dihydroquinolines **1** to act as antioxidants, we studied their behavior by incubating them in different oxidative media [42] (NAD^+^, mouse brain homogenate, H_2_O_2_, riboflavin) (Figure 2). First of all, in vitro stability of the most potent inhibitor **1p** was evaluated in PBS and human plasma at 37 °C. After 3.5 h incubation in these non-oxidizing media, only 3% of the oxidation product **2p** was observed in PBS and 4% in human plasma (8% after 24 h for both media). This finding led us to predict a good stability of **1p** at the periphery. Then, in the presence of the oxidizing medium consisting of 0.2% NAD^+^ in PBS, the oxidation rate of dihydroquinoline **1p** was relatively low (5% after 3.5 h and 16% after 24 h). However, the formation of the oxidation product **2p** was shown to be much faster when using 2% NAD^+^ (24% after 3.5 h and 77% after 24 h). A similar result was observed when **1p** was exposed to 20% fresh mouse brain homogenate (25% after 3.5 h and 72% after 24 h), while hydrogen peroxide 0.1% appeared to be slightly less effective (13% after 3.5 h and 53% after 24 h). Finally, in vitro oxidation with 0.1% riboflavin showed an extremely rapid oxidative conversion of **1p** into the corresponding quaternary salt **2p** (100% in less than 3.5 h).

Finally, the aptitude of **1p** to act as a free radical scavenger was assessed using the rapid, inexpensive and widely used DPPH^•^ (2,2-diphenyl-1-picrylhydrazyl) scavenging method [43]. Dark purple DPPH^•^ is a stable free radical that can accept an electron or hydrogen to become a stable diamagnetic molecule with the loss of violet color. Interestingly, we were able to determine that dihydroquinoline **1p** exhibited a noticeable radical scavenging activity (EC_50_ = 128 μM). Moreover, ^1^H NMR experiments conducted in DMSO-d_6_ showed that dihydroquinoline **1p** reacts with DPPH^•^ almost instantly to furnish the corresponding inactive quinolinium salt **2p** (see Appendix A). Thus, this novel class of *h*DYRK1A/*h*CLK1 inhibitors are all the more interesting that they may also potentially display an additional antioxidant activity to combat oxidative stress in AD.

### 2.4. Drug-likeness Evaluation

Drug candidates should possess favorable ADME (Absorption, Distribution, Metabolism, and Excretion) properties. In the literature, Lipinski [44], based on a study of orally active drugs, established a set of simple rules for estimating permeability. These rules were later adjusted and others were added to design successful CNS drugs [45]. To predict the CNS druggability of **1p**, its ability to cross the blood-brain barrier (BBB) by passive diffusion was examined using the free online server SwissADME [46] (Table 2). According to the results obtained (see Appendix A) and as illustrated by the “BOILED-egg” (Brain Or IntestinaL EstimateD permeation) [47], 1,4-dihydroquinoline **1p** may be viewed as a promising CNS drug candidate. The “BOILED-egg” method is based on a descriptive delineation of well-absorbed drugs. This robust and accurate predictive model of passive absorption is based on the calculation of physicochemical descriptors. 

To confirm these encouraging results, the ability of **1p** to cross the BBB by passive diffusion was assessed through the parallel artificial membrane permeability assay of blood-brain barrier (PAMPA-BBB) [48]. The obtained permeability value suggests that **1p** easily diffuses across the BBB (P_e_ = 20.0 ± 0.9 × 10^−6^ cm·s^−1^).

### 2.5. Docking Studies

Visualization of the *h*DYRK1A structure co-crystallized with INDY (PDB ID: 3ANQ [30]), a DYRK1A benzothiazole derivative ATP-competitive inhibitor (IC_50_ = 0.24 µM and K_i_ = 0.18 µM), and *h*CLK1 co-crystallized with the INDY derivative inhibitor TG003 (PDB ID: 6YTE [49], K_d_ = 75 nM) showed that the two ligands bind in both kinases’ active sites through a very similar interaction network (Figure 3A,B). In both structures, the aromatic ring of the ligands was placed parallel to the beta sheet delimiting the ATP binding site and the ligands established two hydrogen bonds: (i) a first one with a backbone NH of Leu241-*h*DYRK1A or Leu244-*h*CLK1, and (ii) a second one with NH_3_^+^ of the Lys188-*h*DYRK1A or Lys191-*h*CLK1 side chain. Even though the lysine side chain is long and flexible, in both kinases its orientation is fixed through salt bridges with Glu203-*h*DYRK1A and Glu206-*h*CLK1. Therefore, for a ligand binding in the “INDY way” to the two kinases, it is necessary to have two hydrogen bond acceptors at a distance of about 8.5 Å. Among the synthetized ligands, only ligands with the EWG = COCH_3_ and R^1^ substituent = OCH_3_ presented interesting activities. Indeed, the distance between their oxygen atoms is about 8.7 Å. To gain further insight into the binding mode of the dihydroquinoline derivatives, compound **1h** docking studies using the GOLD program applying the ChemPLP scoring function were carried out. GOLD is an automated ligand docking program that uses a genetic algorithm. GOLD’s evolutionary algorithm modifies the position, orientation and conformation of a ligand to fit into one or more low energy states of the protein’s active site. Our docking studies confirmed that compound **1h** reproduces well the INDY binding mode in *h*DYRK1A and in *h*CLK1. For both kinases, the proposed poses by GOLD converged to the same ligand orientation and the best scoring ones are represented on Figure 3C,D. The compound **1h** ChemPLP fit score was 63.59 in *h*DYRK1A and 67.99 in *h*CLK1. As can be seen on Figure 3, compound **1h** is able to establish the two crucial hydrogen bonds with both kinases, the first one with the Leu backbone NH and the second one with NH_3_^+^ of Lys.

## 3. Materials and Methods

### 3.1. Docking Studies

The initial model of compound **1h** was built using BIOVIA Discovery Studio v19.1 (BIOVIA, San Diego, CA, USA) and its preferential protonation state at pH 7.4 was checked using standard tools of the ChemAxon Package (ChemAxon Ltd., Budapest, Hungary) [50]; the majority nonprotonated microspecies was used for the docking studies.

The crystallographic coordinates of *human* CLK1 used in this study were obtained from the X-ray structure of CLK1 bound with the benzothiazole TG003 inhibitor (PDB ID 6YTE [49], a structure refined to 2.30 Å with an R factor of 18.0%) and those of *human* DYRK1A from the X-ray structure of DYRK1A co-crystallized with the INDY inhibitor (PDB ID 3ANQ [30], a structure refined to 2.60 Å with an R factor of 23.6%).

The docking of compound **1h** into the *h*CLK1 and *h*DYRK1 was carried out with the GOLD program v5.3 (The Cambridge Crystallographic Data Centre CCDC, Cambridge, United Kingdom) using the default parameters [51,52]. This program applies a genetic algorithm to explore conformational spaces and ligand binding modes. To evaluate the proposed ligand positions, the ChemPLP fitness function was applied. The binding site in both kinases was defined as a 6 Å sphere from the co-crystallized ligand.

### 3.2. Chemistry

All commercial reagents were used without further purification. The solvents were dried with appropriate desiccants and distilled prior to use or were obtained anhydrous from commercial suppliers. Silica gel (60, 230–400 mesh or 70–230 mesh) was used for column chromatography. Reactions were monitored by thin layer chromatography on silica gel precoated aluminum plates. UV light at 254 nm or KMnO_4_ stains were used to visualize TLC plates. ^1^H, ^13^C, ^19^F NMR spectra were recorded using a spectrometer operating at 300, 75 and 282 MHz, respectively. Abbreviations used for peak multiplicities are s: singlet, d: doublet, t: triplet, q: quadruplet dd = doublet of doublet, br = broad and m: multiplet. Coupling constants *J* are in Hz and chemical shifts are given in ppm and calibrated with DMSO-*d_6_* or CDCl_3_ (residual solvent signals). ^1^H NMR spectra obtained in CDCl_3_ were referenced to 7.26 ppm. ^13^C NMR spectra obtained in CDCl_3_ were referenced to 77.16 ppm and in DMSO-*d*_6_ were referenced to 39.52 ppm. ^19^F NMR chemical shifts (δ) were determined relative to CFCl_3_ as an internal standard (^19^F, δ = 0.0 ppm). Nitrobenzaldehyde derivatives **3b**–**e**, **g**, **h** and 4,4-dimethoxybutan-2-one **4** were obtained from Sigma-Aldrich, while **3f** was commercially available at Fisher Scientific. 7-methoxy quinoline carboxylic acid **5** was prepared as previously reported in the literature [33,35,41]. 3-acetylquinoline **6a** was purchased from Fisher Scientific. 7-methoxyquinoline-3-carbonitrile **6i** was prepared using a synthesis previously described [33]. Elemental analyses were performed by the microanalysis service of the University of Rouen and were recorded with a Thermo Scientific™ FLASH 2000 analyzer (Thermo Fisher Scientific, Waltham, MA, USA).

*General procedure A for the synthesis of compounds***1a**–**p**

The corresponding quinolinium salt **2a**–**p** and BNAH were placed in dry and degassed dichloromethane in a round-bottomed flask and under inert atmosphere. The resulting suspension was stirred in the dark at 20 °C for 4 h. Thereafter, degassed dichloromethane (20 mL) was added to the reaction mixture and the solution was washed with degassed water (2 × 20 mL) and brine. The organic phase was dried on MgSO_4_ and evaporated to dryness to give the corresponding dihydroquinoline.

*1-(1-Methyl-1,4-dihydroquinolin-3-yl)ethanone* (**1a**)

According to procedure A, from quinolinium salt **2a** (150 mg, 0.48 mmol), BNAH (103 mg, 0.48 mmol), dry dichloromethane (2.0 mL). Compound **1a** was obtained (63 mg, 70%) as a brown amorphous solid. ^1^H NMR (300 MHz, CDCl_3_) δ 7.19–7.07 (m, 3H), 6.99 (dd, *J* = 7.4, 1.2 Hz, 1H), 6.78 (dd, *J* = 8.0, 1.1 Hz, 1H), 3.77 (s, 2H), 3.29 (s, 3H), 2.24 (s, 3H); ^13^C NMR (75 MHz, CDCl_3_) δ 194.26, 144.92, 138.45, 129.84, 127.13, 123.82, 123.59, 112.77, 109.45, 39.34, 25.65, 24.12; HRMS (ESI): *m*/*z* calcd. for C_12_H_14_NO: 188.1075 [M + H]^+^, found 188.4068. Anal. Calcd. for C_12_H_13_NO: C, 76.98; H, 7.00; N, 7.48. Found: C, 77.09; H, 7.03; N, 7.53.

*1-(1-Methyl-7-(trifluoromethyl)-1,4-dihydroquinolin-3-yl)ethanone* (**1b**)

According to procedure A, from quinolinium salt **2b** (150 mg, 0.39 mmol), BNAH (84 mg, 0.39 mmol), dry dichloromethane (2.0 mL). Compound **1b** was obtained (66 mg, 66%) as an orange solid. m.p. 141–142 °C; ^1^H NMR (300 MHz, CDCl_3_) δ 7.22–7.12 (m, 3H), 6.93 (d, *J* = 1.6 Hz, 1H), 3.82–3.70 (m, 2H), 3.30 (s, 3H), 2.24 (s, 3H); ^19^F NMR (282 MHz, CDCl_3_) δ −62.7; ^13^C NMR (75 MHz, CDCl_3_) δ 194.30, 144.47, 139.22, 130.22, 129.67 (q, *J* = 32.5 Hz), 127.80 (d, *J* = 1.4 Hz), 124.08 (q, *J* = 1080 Hz) 120.07 (q, *J* = 3.9 Hz), 109.88, 109.31 (q, *J* = 3.9 Hz), 39.38, 25.69, 24.23; HRMS (ESI): *m*/*z* calcd. for C_13_H_13_F_3_NO: 256.0949 [M + H]^+^, found 256.0955. Anal. Calcd. for C_13_H_12_F_3_NO: C, 61.17; H, 4.74; N, 5.49. Found: C, 61.09; H, 4.83; N, 5.53.

*1-(7-(Dimethylamino)-1-methyl-1,4-dihydroquinolin-3-yl)ethanone* (**1c**)

According to procedure A, from quinolinium salt **2c** (150 mg, 0.42 mmol), BNAH (90 mg, 0.42 mmol), dry dichloromethane (2.0 mL). Compound **1c** was obtained (65 mg, 67%) as an orange solid. m.p. 126–127 °C; ^1^H NMR (300 MHz, CDCl_3_) δ 7.14 (d, *J* = 0.8 Hz, 1H), 6.94 (d, *J* = 8.4 Hz, 1H), 6.38 (dd, *J* = 8.3, 2.5 Hz, 1H), 6.12 (d, *J* = 2.4 Hz, 1H), 3.65 (s, 2H), 3.28 (s, 3H), 2.91 (s, 6H), 2.21 (s, 3H); ^13^C NMR (75 MHz, CDCl_3_) δ 194.45, 150.19, 145.12, 138.88, 130.23, 111.92, 110.02, 108.19, 97.92, 40.88, 39.41, 24.77, 24.16; HRMS (ESI): *m*/*z* calcd. for C_14_H_19_N_2_O: 231.1497 [M + H]^+^, found 231.1494. Anal. Calcd. for C_14_H_18_N_2_O: C, 73.01; H, 7.88; N, 12.16. Found: C, 72.89; H, 7.82; N, 12.23.

*1-(7-Bromo-1-methyl-1,4-dihydroquinolin-3-yl)ethanone* (**1d**)

According to procedure A, from quinolinium salt **2d** (150 mg, 0.38 mmol), BNAH (82 mg, 0.38 mmol), dry dichloromethane (2.0 mL). Compound **1d** was obtained (75 mg, 74%) as a beige solid. m.p. 179–180 °C; ^1^H NMR (300 MHz, CDCl_3_) δ 7.13–7.05 (m, 2H), 6.95 (dt, *J* = 8.1, 1.1 Hz, 1H), 6.89 (d, *J* = 1.9 Hz, 1H), 3.69 (s, 2H), 3.26 (s, 3H), 2.24 (s, 3H); ^13^C NMR (75 MHz, CDCl_3_) δ 194.33, 144.39, 139.93, 131.10, 126.16, 122.72, 120.46, 115.77, 110.09, 39.40, 25.27, 24.25; HRMS (ESI): *m*/*z* calcd. for C_12_H_13_NO^79^Br: 266.0181 [M + H]^+^, found 266.0182. Anal. Calcd. for C_12_H_12_NOBr: C, 54.16; H, 4.54; N, 5.26. Found: C, 54.06; H, 4.67; N, 5.23.

*1-(6-Chloro-1-methyl-1,4-dihydroquinolin-3-yl)ethanone* (**1e**)

According to procedure A, from quinolinium salt **2e** (150 mg, 0.43 mmol), BNAH (93 mg, 0.43 mmol), dry dichloromethane (2.0 mL). Compound **1e** was obtained (74 mg, 77%) as a beige solid. m.p. 124–125 °C; ^1^H NMR (300 MHz, CDCl_3_) δ 7.14–7.05 (m, 3H), 6.72–6.65 (m, 1H), 3.73 (d, *J* = 1.1 Hz, 2H), 3.27 (s, 3H), 2.23 (s, 3H); ^13^C NMR (75 MHz, CDCl_3_) δ 194.10, 144.52, 137.17, 129.45, 128.34, 126.92, 125.63, 113.91, 109.22, 39.41, 25.60, 24.13; HRMS (ESI): *m*/*z* calcd. for C_12_H_13_NO^35^Cl: 222.0686 [M + H]^+^, found 222.0692. Anal. Calcd. for C_12_H_12_NOCl: C, 65.02; H, 5.46; N, 6.32. Found: C, 64.97; H, 5.53; N, 6.22.

*1-(6,7-Dimethoxy-1-methyl-1,4-dihydroquinolin-3-yl)ethanone* (**1f**)

According to procedure A, from quinolinium salt **2f** (160 mg, 0.43 mmol), BNAH (92 mg, 0.43 mmol), dry dichloromethane (2.0 mL). Compound **1f** was obtained (10 mg, 10%) as a yellow solid. m.p. 235–236 °C; ^1^H NMR (300 MHz, CDCl_3_) δ 7.14 (s, 1H), 6.62 (t, *J* = 0.9 Hz, 1H), 6.38 (s, 1H), 3.87 (s, 3H), 3.83 (s, 3H), 3.70 (s, 2H), 3.30 (s, 3H), 2.22 (s, 3H); ^13^C NMR (75 MHz, CDCl_3_) δ 194.29, 147.87, 145.58, 144.72, 131.86, 115.79, 113.28, 108.77, 98.74, 56.49, 56.36, 39.63, 25.62, 24.21; HRMS (ESI): *m*/*z* calcd. for C_14_H_18_NO_3_: 248.1287 [M + H]^+^, found 248.1282. Anal. Calcd. for C_14_H_17_NO_3_: C, 68.00; H, 6.93; N, 5.66. Found: C, 67.91; H, 6.83; N, 5.73.

*1-(5-Methyl-5,8-dihydro-*[1,3]*dioxolo[4,5-g]quinolin-7-yl)ethanone* (**1g**)

According to procedure A, from quinolinium salt **2g** (150 mg, 0.42 mmol), BNAH (90 mg, 0.42 mmol), dry dichloromethane (2.0 mL). Compound **1g** was obtained (65 mg, 67%) as an orange solid. m.p. 152–153 °C; ^1^H NMR (300 MHz, CDCl_3_) δ 7.09 (s, 1H), 6.55 (t, *J* = 0.9 Hz, 1H), 6.36 (s, 1H), 5.87 (s, 2H), 3.64 (s, 2H), 3.22 (s, 3H), 2.19 (s, 3H); ^13^C NMR (75 MHz, CDCl_3_) δ 194.16, 146.71, 144.66, 143.60, 132.73, 116.50, 109.61, 108.49, 101.23, 95.40, 39.88, 25.97, 24.12; HRMS (ESI): *m*/*z* calcd. for C_13_H_14_NO_3_: 232.0974 [M + H]^+^, found 232.0975. Anal. Calcd. for C_13_H_13_NO_3_: C, 67.52; H, 5.67; N, 6.06. Found: C, 67.72; H, 5.63; N, 6.11.

*1-(7-Methoxy-1-methyl-1,4-dihydroquinolin-3-yl)ethanone* (**1h**)

According to procedure A, from quinolinium salt **2h** (138 mg, 0.47 mmol), BNAH (100 mg, 0.47 mmol), dry dichloromethane (2.0 mL). Compound **1h** was obtained (83 mg, 82%) as an orange solid. m.p. 99–100 °C; ^1^H NMR (300 MHz, CDCl_3_) δ 7.14 (d, *J* = 0.8 Hz, 1H), 7.02 (dt, *J* = 8.3, 1.1 Hz, 1H), 6.53 (dd, *J* = 8.3, 2.5 Hz, 1H), 6.35 (d, *J* = 2.4 Hz, 1H), 3.79 (s, 3H), 3.69 (s, 2H), 3.27 (s, 3H), 2.24 (s, 3H); ^13^C NMR (75 MHz, CDCl_3_) δ 194.54, 158.97, 144.79, 139.42, 130.46, 116.12, 110.25, 107.36, 100.48, 55.54, 39.52, 24.93, 24.29; HRMS (ESI): *m*/*z* calcd. for C_13_H_16_NO_2_: 218.1181 [M + H]^+^, found 218.1191. Anal. Calcd. for C_13_H_15_NO_2_: C, 71.87; H, 6.96; N, 6.45. Found: C, 71.91; H, 6.93; N, 6.53.

*7-Methoxy-1-methyl-1,4-dihydroquinoline-3-carbonitrile* (**1i**)

According to procedure A, from quinolinium salt **2i** (100 mg, 0.31 mmol), BNAH (66 mg, 0.31 mmol), dry dichloromethane (2.0 mL). Compound **1i** was obtained (45 mg, 73%) as a yellow solid. m.p. 127–128 °C; ^1^H NMR (300 MHz, CDCl_3_) δ 6.88 (dt, *J* = 8.3, 1.0 Hz, 1H), 6.66 (t, *J* = 1.0 Hz, 1H), 6.51 (dd, *J* = 8.3, 2.5 Hz, 1H), 6.29 (d, *J* = 2.4 Hz, 1H), 3.78 (s, 3H), 3.64 (d, *J* = 1.1 Hz, 2H), 3.15 (s, 3H); ^13^C NMR (75 MHz, CDCl_3_) δ 159.33, 144.92, 139.15, 129.87, 121.41, 112.54, 107.50, 100.36, 77.51, 55.47, 38.95, 26.68; HRMS (ESI): *m*/*z* calcd. for C_12_H_13_N_2_O: 201.1028 [M + H]^+^, found 201.1035. Anal. Calcd. for C_12_H_12_N_2_O: C, 71.98; H, 6.04; N, 13.99. Found: C, 72.05; H, 5.98; N, 14.08.

*Methyl 7-methoxy-1-methyl-1,4-dihydroquinoline-3-carboxylate* (**1j**)

According to procedure A, from quinolinium salt **2j** (100 mg, 0.28 mmol), BNAH (60 mg, 0.28 mmol), dry dichloromethane (2.0 mL). Compound **1j** was obtained (51 mg, 79%) as a yellow solid. m.p. 95–96 °C; ^1^H NMR (300 MHz, CDCl_3_) δ 7.22 (d, *J* = 0.9 Hz, 1H), 6.97 (dt, *J* = 8.3, 1.1 Hz, 1H), 6.50 (dd, *J* = 8.3, 2.5 Hz, 1H), 6.31 (d, *J* = 2.4 Hz, 1H), 3.79 (s, 3H), 3.72 (d, *J* = 6.7 Hz, 5H), 3.20 (s, 3H); ^13^C NMR (75 MHz, CDCl_3_) δ 168.37, 159.00, 143.10, 139.91, 130.24, 115.39, 106.95, 100.20, 97.69, 55.46, 51.13, 39.05, 25.56; HRMS (ESI): *m*/*z* calcd. for C_13_H_16_NO_3_: 234.1130 [M + H]^+^, found 234.1135. Anal. Calcd. for C_13_H_15_NO_3_: C, 66.94; H, 6.48; N, 6.00. Found: C, 67.03; H, 6.53; N, 6.09.

*Ethyl 7-methoxy-1-methyl-1,4-dihydroquinoline-3-carboxylate* (**1k**)

According to procedure A, from quinolinium salt **2k** (100 mg, 0.27 mmol), BNAH (58 mg, 0.27 mmol), dry dichloromethane (2.0 mL). Compound **1k** was obtained (57 mg, 86%) as a yellow solid. m.p. 93–94 °C; ^1^H NMR (300 MHz, CDCl_3_) δ 7.20 (d, *J* = 0.9 Hz, 1H), 6.96 (dt, *J* = 8.3, 1.1 Hz, 1H), 6.49 (dd, *J* = 8.3, 2.5 Hz, 1H), 6.29 (d, *J* = 2.5 Hz, 1H), 4.18 (q, *J* = 7.1 Hz, 2H), 3.78 (s, 3H), 3.69 (d, *J* = 1.1 Hz, 2H), 3.19 (s, 3H), 1.29 (t, *J* = 7.1 Hz, 3H); ^13^C NMR (75 MHz, CDCl_3_) δ 167.96, 158.99, 142.88, 139.99, 130.22, 115.45, 106.87, 100.14, 98.03, 59.66, 55.46, 39.03, 25.56, 14.69; HRMS (ESI): *m*/*z* calcd. for C_14_H_18_NO_3_: 248.1287 [M + H]^+^, found 248.1279. Anal. Calcd. for C_14_H_17_NO_3_: C, 68.00; H, 6.93; N, 5.66. Found: C, 68.05; H, 6.99; N, 5.59.

*7-Methoxy-N,N,1-trimethyl-1,4-dihydroquinoline-3-carboxamide* (**1l**)

According to procedure A, from quinolinium salt **2l** (150 mg, 0.40 mmol), BNAH (86 mg, 0.40 mmol), dry dichloromethane (2.0 mL). Compound **1l** was obtained (59.7 mg, 60%) as a yellow solid. m.p. 219–220 °C; ^1^H NMR (300 MHz, CDCl_3_) δ 6.92 (dt, *J* = 8.3, 1.0 Hz, 1H), 6.48–6.39 (m, 2H), 6.25 (d, *J* = 2.5 Hz, 1H), 3.76 (s, 3H), 3.65 (d, *J* = 1.1 Hz, 2H), 3.11 (s, 3H), 3.01 (s, 6H); ^13^C NMR (75 MHz, CDCl_3_) δ 171.95, 159.04, 141.19, 138.15, 129.58, 114.53, 105.96, 102.37, 99.17, 55.39, 38.49, 37.79, 27.75; HRMS (ESI): *m*/*z* calcd. for C_14_H_19_N_2_O_2_: 247.1447 [M + H]^+^, found 247.1457. Anal. Calcd. For C_14_H_18_N_2_O_2_: C, 68.27; H, 7.37; N, 11.37. Found: C, 68.07; H, 7.18; N, 11.29.

*N,7-Dimethoxy-N,1-dimethyl-1,4-dihydroquinoline-3-carboxamide* (**1m**)

According to procedure A, from quinolinium salt **2m** (150 mg, 0.39 mmol), BNAH (83 mg, 0.39 mmol), dry dichloromethane (3.0 mL). Compound **1m** was obtained (68 mg, 67%) as a yellow solid. ^1^H NMR (300 MHz, CDCl_3_) δ 7.23 (d, *J* = 0.9 Hz, 1H), 6.95 (dt, *J* = 8.2, 1.0 Hz, 1H), 6.47 (dd, *J* = 8.3, 2.5 Hz, 1H), 6.28 (d, *J* = 2.4 Hz, 1H), 3.77 (d, *J* = 3.1 Hz, 5H), 3.64 (s, 3H), 3.21 (s, 3H), 3.17 (s, 3H); ^13^C NMR (75 MHz, CDCl_3_) δ 169.90, 158.96, 142.60, 140.41, 129.78, 115.72, 106.51, 100.47, 99.61, 60.38, 55.43, 38.95, 34.17, 27.13; HRMS (ESI): *m*/*z* calcd. For C_14_H_19_NO_3_: 263.1396 [M + H]^+^, found 263.1397. Anal. Calcd. for C_14_H_18_N_2_O_3_: C, 64.10; H, 6.92; N, 10.68. Found: C, 64.19; H, 6.90; N, 10.73.

*1-(1-Benzyl-7-methoxy-1,4-dihydroquinolin-3-yl)ethanone* (**1n**)

According to procedure A, from quinolinium salt **2n** (160 mg, 0.43 mmol), BNAH (92 mg, 0.43 mmol), dry dichloromethane (2.0 mL). Compound **1n** was obtained (86 mg, 68%) as an orange solid. m.p. 129–130 °C; ^1^H NMR (300 MHz, CDCl_3_) δ 7.39–7.25 (m, 6H), 7.01 (dt, *J* = 8.3, 1.1 Hz, 1H), 6.47 (dd, *J* = 8.3, 2.4 Hz, 1H), 6.24 (d, *J* = 2.4 Hz, 1H), 4.80 (s, 2H), 3.75 (s, 2H), 3.63 (s, 3H), 2.23 (s, 3H); ^13^C NMR (75 MHz, CDCl_3_) δ 194.75, 158.74, 144.41, 138.41, 136.25, 130.59, 129.11, 127.83, 126.18, 116.02, 110.79, 107.70, 101.31, 55.32, 55.27, 25.00, 24.28; HRMS (ESI): *m*/*z* calcd. for C_19_H_20_NO_2_: 294.1494 [M + H]^+^, found 294.1490. Anal. Calcd. for C_19_H_19_NO_2_: C, 77.79; H, 6.53; N, 4.77. Found: C, 77.86; H, 6.50; N, 4.70.

*1-(7-Methoxy-1-phenethyl-1,4-dihydroquinolin-3-yl)ethanone* (**1o**)

According to procedure A, from quinolinium salt **2o** (150 mg, 0.39 mmol), BNAH (83 mg, 0.39 mmol), dry dichloromethane (2.0 mL). Compound **1o** was obtained (90 mg, 75%) as an orange solid. m.p. 92–93 °C; ^1^H NMR (300 MHz, CDCl_3_) δ 7.29 (dd, *J* = 6.9, 1.3 Hz, 2H), 7.26–7.13 (m, 3H), 7.02 (dt, *J* = 8.3, 1.1 Hz, 1H), 6.72 (s, 1H), 6.53 (dd, *J* = 8.3, 2.4 Hz, 1H), 6.47 (d, *J* = 2.4 Hz, 1H), 3.80 (s, 5H), 3.62 (s, 2H), 2.99 (t, *J* = 6.9 Hz, 2H), 1.99 (s, 3H); ^13^C NMR (75 MHz, CDCl_3_) δ 194.63, 158.94, 144.23, 138.13, 137.87, 130.95, 129.05, 128.86, 126.97, 116.40, 109.69, 107.23, 100.53, 55.47, 53.25, 34.24, 24.76, 23.97; HRMS (ESI): *m*/*z* calcd. for C_20_H_22_NO_2_: 308.1651 [M + H]^+^, found 308.1647. Anal. Calcd. for C_20_H_21_NO_2_: C, 78.15; H, 6.89; N, 4.56. Found: C, 78.25; H, 6.91; N, 4.49.

*1-(7-Methoxy-1-propyl-1,4-dihydroquinolin-3-yl)ethanone* (**1p**)

According to procedure A, from quinolinium salt **2p** (130 mg, 0.40 mmol), BNAH (86 mg, 0.40 mmol), dry dichloromethane (2.0 mL). Compound **1p** was obtained (60 mg, 61%) as an orange solid. m.p. 98–99 °C; ^1^H NMR (300 MHz, CDCl_3_) δ 7.15 (d, *J* = 0.8 Hz, 1H), 7.01 (dt, *J* = 8.4, 1.1 Hz, 1H), 6.50 (dd, *J* = 8.3, 2.4 Hz, 1H), 6.35 (d, *J* = 2.4 Hz, 1H), 3.78 (s, 3H), 3.68 (s, 2H), 3.59–3.47 (m, 2H), 2.23 (s, 3H), 1.82–1.69 (m, 2H), 0.99 (t, *J* = 7.4 Hz, 3H); ^13^C NMR (75 MHz, CDCl_3_) δ 194.59, 158.92, 144.22, 138.25, 130.82, 116.46, 110.00, 107.10, 100.82, 55.53, 53.60, 25.07, 24.29, 21.38, 11.33; HRMS (ESI): *m*/*z* calcd. for C_15_H_20_NO_2_: 246.1494 [M + H]^+^, found 246.1507. Anal. Calcd. for C_15_H_19_NO_2_: C, 73.44; H, 7.81; N, 5.71. Found: C, 73.46; H, 7.80; N, 5.80.

*General procedure B for the synthesis of compounds***2a**–**p**

In a glass tube, to a solution of the corresponding quinoline **6a**–**m** in dry acetonitrile was added a large excess of alkyl halide at 20 °C. The tube was sealed and the resulting mixture was heated at 85 °C until reaction completion (18 h for MeI and up to 4 days for the rest of the alkylating agents). After cooling at room temperature, dichloromethane (5 mL) and diethyl ether (15 mL) were added to the suspension and the mixture was stirred for 10 min. The formed precipitate was filtered, rinsed twice with diethyl ether, and dried under vacuum to afford the corresponding quinolinium salt.

*3-Acetyl-1-methylquinolin-1-ium iodide* (**2a**)

According to procedure B, from commercially available 3-acetylquinoline (200 mg, 1.17 mmol), MeI (1.0 mL), dry acetonitrile (2.0 mL). Quinolinium salt **2a** (340 mg, 93%) was obtained as an orange powder. m.p. 235–236 °C; ^1^H NMR (300 MHz, DMSO) δ 9.97 (s, 1H), 9.90 (s, 1H), 8.79–8.53 (m, 2H), 8.42 (td, *J* = 7.2, 3.5 Hz, 1H), 8.16 (t, *J* = 7.6 Hz, 1H), 4.73 (s, 3H), 2.84 (s, 3H); ^13^C NMR (75 MHz, DMSO) δ 194.42, 149.89, 146.97, 139.08, 137.56, 132.05, 130.78, 129.54, 128.43, 119.50, 45.64, 27.26; HRMS (ESI): *m*/*z* calcd. for C_12_H_12_NO: 186.0919 [M-I]^+^, found 186.0916.

*3-Acetyl-1-methyl-7-(trifluoromethyl)quinolin-1-ium iodide* (**2b**)

According to procedure B, from quinoline **6b** (200 mg, 0.43 mmol), MeI (1.0 mL), dry acetonitrile (4.0 mL). Quinolinium salt **2b** (261 mg, 82%) was obtained as a reddish-orange powder. m.p. 227–228 °C; ^1^H NMR (300 MHz, DMSO) δ 10.10 (dd, *J* = 1.9, 0.8 Hz, 1H), 9.99 (s, 1H), 8.93 (s, 1H), 8.86 (d, *J* = 8.6 Hz, 1H), 8.48 (dd, *J* = 8.5, 1.5 Hz, 1H), 4.83 (s, 3H), 2.86 (s, 3H); ^13^C NMR (75 MHz, CDCl_3_) δ 196.34, 150.43, 148.74, 136.92, 133.26 (q, *J* = 32.9 Hz), 130.55, 129.01, 128.34, 127.19 (q, *J* = 4.4 Hz), 125.40, 123.20 (q, *J* = 3.1 Hz), 121.78, 26.86; ^19^F NMR (282 MHz, CDCl_3_) δ −61.35; HRMS (ESI): *m*/*z* calcd. for C_13_H_11_NOF_3_: 254.0793 [M-I]^+^, found 254.0798. Anal. Calcd. for C_13_H_11_NOF_3_I: C, 40.97; H, 2.91; N, 3.68. Found: C, 40.82; H, 2.98; N, 3.75.

*3-Acetyl-7-dimethylamino-1-methylquinolin-1-ium iodide* (**2c**)

According to procedure B, from quinoline **6c** (200 mg, 0.93 mmol), MeI (1.0 mL), dry acetonitrile (4.0 mL). Quinolinium salt **2c** (304 mg, 91%) was obtained as a brick orange powder. m.p. 279–280 °C; ^1^H NMR (300 MHz, DMSO) δ 9.48 (d, *J* = 1.9 Hz, 1H), 9.23 (d, *J* = 1.5 Hz, 1H), 8.26 (d, *J* = 9.5 Hz, 1H), 7.66 (dd, *J* = 9.4, 2.3 Hz, 1H), 6.93–6.78 (m, 1H), 4.38 (s, 3H), 3.33 (s, 6H), 2.69 (s, 3H); ^13^C NMR (75 MHz, DMSO) δ 193.93, 155.70, 148.17, 142.46, 142.04, 133.36, 123.36, 121.26, 119.32, 94.15, 44.21, 40.53, 26.77; HRMS (ESI): *m*/*z* calcd. for C_14_H_17_N_2_O: 229.1341 [M-I]^+^, found 229.1337. Anal. Calcd. for C_14_H_17_N_2_OI: C, 47.21; H, 4.81; N, 7.86. Found: C, 47.11; H, 4.78; N, 7.77.

*3-Acetyl-7-bromo-1-methylquinolin-1-ium iodide* (**2d**)

According to procedure B, from quinoline **6d** (170 mg, 0.680 mmol), methyl iodide (1.0 mL), dry acetonitrile (2.0 mL). Quinolinium salt **2d** (240 mg, 90%) was obtained as a deep yellow powder. m.p. 248–249 °C; ^1^H NMR (300 MHz, DMSO) δ 9.97 (dd, *J* = 1.9, 0.8 Hz, 1H), 9.87 (d, *J* = 1.8 Hz, 1H), 8.89 (t, *J* = 1.2 Hz, 1H), 8.56 (d, *J* = 8.7 Hz, 1H), 8.35 (dd, *J* = 8.7, 1.7 Hz, 1H), 4.70 (s, 3H), 2.82 (s, 3H); ^13^C NMR (75 MHz, DMSO) δ 194.19, 150.65, 146.82, 139.56, 134.20, 133.38, 132.10, 129.76, 127.40, 122.40, 45.77, 27.18; HRMS (ESI): *m*/*z* calcd. for C_12_H_11_NO^79^Br: 264.0024 [M-I]^+^, found 264.0023. Anal. Calcd. for C_12_H_11_NOBrI: C, 36.76; H, 2.83; N, 3.57. Found: C, 36.70; H, 2.78; N, 3.49.

*3-Acetyl-6-chloro-1-methylquinolin-1-ium iodide* (**2e**)

According to procedure B, from quinoline **6e** (200 mg, 0.973 mmol), methyl iodide (1.0 mL), dry acetonitrile (2.0 mL). Quinolinium salt **2e** (321 mg, 95%) was obtained as a bright orange powder. m.p. 259–260 °C; ^1^H NMR (300 MHz, DMSO) δ 10.00 (dd, *J* = 1.9, 0.8 Hz, 1H), 9.84–9.77 (m, 1H), 8.79 (d, *J* = 2.4 Hz, 1H), 8.62 (d, *J* = 9.3 Hz, 1H), 8.45 (dd, *J* = 9.4, 2.4 Hz, 1H), 4.73 (s, 3H); ^13^C NMR (75 MHz, DMSO) δ 194.05, 150.34, 145.92, 137.88, 137.22, 135.21, 130.53, 130.24, 129.41, 121.89, 45.85, 27.17; HRMS (ESI): *m*/*z* calcd. for C_12_H_11_NOCl: 220.0529 [M-I]^+^, found 220.0536. Anal. Calcd. for C_12_H_11_NOClI: C, 41.47; H, 3.19; N, 4.03. Found: C, 41.50; H, 3.28; N, 3.97.

*3-Acetyl-6,7-dimethoxy-1-methylquinolin-1-ium iodide* (**2f**)

According to procedure B, from quinoline **6f** (200 mg, 0.87 mmol), MeI (1.0 mL), dry acetonitrile (4.0 mL). Quinolinium salt **2f** (290 mg, 90%) was obtained as a yellow powder. m.p. 235–236°C; ^1^H NMR (300 MHz, DMSO) δ 9.71–9.64 (m, 1H), 9.53 (d, *J* = 1.8 Hz, 1H), 8.04 (s, 1H), 7.71 (s, 1H), 4.66 (s, 3H), 4.20 (s, 3H), 4.03 (s, 3H), 2.79 (s, 3H); ^13^C NMR (75 MHz, DMSO) δ 194.21, 158.23, 151.48, 145.88, 142.81, 137.42, 127.55, 125.03, 108.95, 99.19, 57.66, 56.64, 45.56, 27.08; HRMS (ESI): *m*/*z* calcd. for C_14_H_16_NO_3_: 246.1130 [M-I]^+^, found 246.1138. Anal. Calcd. for C_14_H_16_NO_3_I: C, 45.06; H, 4.32; N, 3.75. Found: C, 45.11; H, 4.46; N, 3.78.

*7-Acetyl-5-methyl-*[1,3]*dioxolo[4,5-g]quinolin-5-ium iodide* (**2g**)

According to procedure B, from quinoline **6g** (150 mg, 0.745 mmol), methyl iodide (1.0 mL), dry acetonitrile (4.0 mL). Quinolinium salt **2g** (222 mg, 89%) was obtained as a beige powder. m.p. 235–236 °C; ^1^H NMR (300 MHz, DMSO) δ 9.70–9.63 (m, 1H), 9.47 (d, *J* = 1.8 Hz, 1H), 8.06 (d, *J* = 0.8 Hz, 1H), 7.94 (s, 1H), 6.54 (s, 2H), 4.56 (s, 3H), 2.77 (s, 3H); ^13^C NMR (75 MHz, DMSO) δ 194.23, 157.51, 150.50, 146.24, 143.30, 139.54, 127.89, 126.93, 105.65, 105.15, 97.21, 45.98, 27.11; HRMS (ESI): *m*/*z* calcd. for C_13_H_12_NO_3_: 230.0817 [M-I]^+^, found 230.0823. Anal. Calcd. for C_13_H_12_NO_3_I: C, 43.72; H, 3.39; N, 3.92. Found: C, 43.77; H, 3.29; N, 3.93.

*3-Acetyl-7-methoxy-1-methylquinolin-1-ium iodide* (**2h**)

According to procedure B, from quinoline **6h** (58 mg, 0.29 mmol), MeI (0.5 mL), dry acetonitrile (1.0 mL). Quinolinium salt **2h** (70 mg, 70%) was obtained as an orange powder. m.p. 223–224 °C; ^1^H NMR (300 MHz, DMSO) δ 9.83 (d, *J* = 1.8 Hz, 1H), 9.70 (d, *J* = 1.8 Hz, 1H), 8.55 (d, *J* = 9.1 Hz, 1H), 7.84–7.71 (m, 2H), 4.64 (s, 3H), 4.17 (s, 3H), 2.79 (s, 3H); ^13^C NMR (75 MHz, DMSO) δ 194.19, 166.59, 149.27, 145.50, 141.93, 133.88, 127.15, 123.90, 122.76, 99.62, 57.32, 45.37, 27.05; HRMS (ESI): *m*/*z* calcd. for C_13_H_14_NO_2_: 216.1025 [M-I]^+^, found 216.1015. Anal. Calcd. for C_13_H_14_NO_2_I: C, 45.50; H, 4.11; N, 4.08. Found: C, 45.50; H, 4.17; N, 4.13.

*3-Cyano-7-methoxy-1-methylquinolin-1-ium* (**2i**)

According to procedure B, from quinoline **6i** (100 mg, 0.54 mmol), MeI (0.5 mL), dry acetonitrile (2.0 mL). Quinolinium salt **2i** (133 mg, 75%) was obtained as an orange powder. m.p. 223–224 °C; ^1^H NMR (300 MHz, DMSO) δ 9.95 (s, 1H), 9.68 (s, 1H), 8.42 (d, *J* = 9.2 Hz, 1H), 7.91–7.65 (m, 2H), 4.55 (s, 3H), 4.18 (s, 3H); ^13^C NMR (75 MHz, DMSO) δ 167.38, 151.67, 149.64, 141.92, 133.29, 123.83, 123.48, 114.89, 102.97, 99.79, 57.62, 45.74; HRMS (ESI): *m*/*z* calcd. for C_12_H_11_N_2_O: 199.0875 [M-I]^+^, found 199.0871.

*7-Methoxy-3-(methoxycarbonyl)-1-methylquinolin-1-ium* (**2j**)

According to procedure B, from quinoline **6j** (100 mg, 0.46 mmol), MeI (0.5 mL), dry acetonitrile (2.0 mL). Quinolinium salt **2j** (141 mg, 85%) was obtained as a deep yellow powder. m.p. 240−241 °C; ^1^H NMR (300 MHz, DMSO) δ 9.85 (d, *J* = 1.8 Hz, 1H), 9.67 (d, *J* = 1.9 Hz, 1H), 8.59 (d, *J* = 9.1 Hz, 1H), 7.90–7.64 (m, 2H), 4.63 (s, 3H), 4.16 (s, 3H), 4.02 (s, 3H); ^13^C NMR (75 MHz, DMSO) δ 166.79, 162.88, 149.66, 146.27, 142.35, 133.81, 124.06, 122.89, 120.77, 99.55, 57.40, 53.31, 45.36; HRMS (ESI): *m*/*z* calcd. for C_13_H_14_NO_3_: 232.0974 [M-I]^+^, found 232.0979. Anal. Calcd. for C_13_H_14_NO_3_I: C, 43.47; H, 3.93; N, 3.90. Found: C, 43.60; H, 3.90; N, 3.90.

*3-(Ethoxycarbonyl)- 7-methoxy-1-methylquinolin-1-ium* (**2k**)

According to procedure B, from quinoline **6k** (100 mg, 0.43 mmol), MeI (0.5 mL), dry acetonitrile (2.0 mL). Quinolinium salt **2k** (161 mg, 84%) was obtained as a deep yellow powder. m.p. 218−219 °C; ^1^H NMR (300 MHz, DMSO) δ 9.83 (d, *J* = 1.9 Hz, 1H), 9.66 (d, *J* = 2.2 Hz, 1H), 8.59 (d, *J* = 9.1 Hz, 1H), 7.79 (dd, *J* = 9.0, 2.2 Hz, 1H), 7.74 (d, *J* = 2.3 Hz, 1H), 4.63 (s, 3H), 4.49 (q, *J* = 7.1 Hz, 2H), 4.16 (s, 3H), 1.41 (t, *J* = 7.1 Hz, 3H);^13^C NMR (75 MHz, DMSO) δ 166.76, 162.38, 149.65, 146.20, 142.35, 133.81, 124.07, 122.87, 120.99, 99.54, 62.34, 57.38, 45.38, 14.21; HRMS (ESI): *m*/*z* calcd. for C_14_H_16_NO_3_: 246.1130 [M-I]^+^, found 246.1235. Anal. Calcd. For C_14_H_16_NO_3_I: C, 45.06; H, 4.32; N, 3.75. Found: C, 45.11; H, 4.46; N, 3.78.

*3-(Dimethylcarbamoyl)-7-methoxy-1-methylquinolin-1-ium iodide* (**2l**)

According to procedure B, from quinoline **6l** (200 mg, 0.87 mmol), MeI (1.0 mL), dry acetonitrile (4.0 mL). Quinolinium salt **2l** (278 mg, 91%) was obtained as a khaki powder. M.p. 219–220 °C; ^1^H NMR (300 MHz, DMSO) δ 9.56 (d, *J* = 1.8 Hz, 1H), 9.28 (d, *J* = 1.8 Hz, 1H), 8.42 (d, *J* = 9.0 Hz, 1H), 7.81–7.65 (m, 2H), 4.59 (s, 3H), 4.15 (s, 3H), 3.10 (s, 6H). ^13^C NMR (75 MHz, DMSO) δ 165.48, 164.57, 148.25, 143.75, 140.93, 132.60, 126.91, 124.00, 122.64, 98.98, 57.23, 45.38, 35.34; HRMS (ESI): *m*/*z* calcd. For C_14_H_17_N_2_O_2_: 245.1290 [M-I]^+^, found 245.1282. Anal. Calcd. for C_14_H_17_N_2_O_2_I: C, 45.18; H, 4.60; N, 7.53. Found: C, 45.03; H, 4.66; N, 7.48.

*7-Methoxy-3-(methoxy(methyl)carbamoyl)-1-methylquinolin-1-ium iodide* (**2m**)

According to procedure B, from quinoline **6m** (350 mg, 0.745 mmol), MeI (3.0 mL), no solvent, 24 h. Quinolinium salt **2m** (471 mg, 85%) was obtained as a khaki powder. m.p. 206–207 °C; ^1^H NMR (300 MHz, DMSO) δ 9.63 (d, *J* = 1.8 Hz, 1H), 9.43 (d, *J* = 1.8 Hz, 1H), 8.51 (d, *J* = 9.1 Hz, 1H), 7.84–7.65 (m, 2H), 4.61 (s, 3H), 4.15 (s, 3H), 3.65 (s, 3H), 3.41 (s, 3H), 3.34 (s, 3H); ^13^C NMR (75 MHz, DMSO) δ 166.00, 162.95, 149.04, 145.11, 141.35, 133.22, 124.38, 123.83, 122.69, 99.21, 61.59, 57.28, 45.50; HRMS (ESI): *m*/*z* calcd. for C_14_H_17_N_2_O_3_: 261.1239 [M-I]^+^, found 261.1243. Anal. Calcd. for C_14_H_17_N_2_O_3_I: C, 43.32; H, 4.41; N, 7.22. Found: C, 43.47; H, 4.46; N, 7.17.

*3-Acetyl-1-benzyl-7-methoxyquinolin-1-ium bromide* (**2n**)

According to procedure B, from quinoline **6h** (150 mg, 0.745 mmol), Benzyl bromide (1.0 mL), dry acetonitrile (4.0 mL), 3 days. Quinolinium salt **2n** (252 mg, 91%) was obtained as a beige powder. m.p. 218–219 °C; ^1^H NMR (300 MHz, DMSO) δ 10.10 (d, *J* = 1.9 Hz, 1H), 9.79 (d, *J* = 1.8 Hz, 1H), 8.57 (d, *J* = 8.9 Hz, 1H), 7.80–7.65 (m, 2H), 7.54–7.29 (m, 5H), 6.44 (s, 2H), 4.00 (s, 3H), 2.84 (s, 3H); ^13^C NMR (75 MHz, DMSO) δ 194.25, 166.49, 149.56, 146.28, 140.94, 134.41, 133.63, 129.10, 128.86, 127.76, 127.65, 124.68, 122.56, 100.21, 59.97, 57.19, 27.16; HRMS (ESI): *m*/*z* calcd. for C_19_H_18_NO_2_: 292.1338 [M-Br]^+^, found 292.1336. Anal. Calcd. for C_19_H_18_NO_2_Br: C, 61.30; H, 4.87; N, 3.76. Found: C, 61.41; H, 4.95; N, 3.79.

*3-Acetyl-7-methoxy-1-phenethylquinolin-1-ium bromide* (**2o**)

According to procedure B, from quinoline **6h** (150 mg, 0.745 mmol), phenethyl bromide (1.0 mL), dry acetonitrile (4.0 mL), 4 days. Quinolinium salt **2o** (230 mg, 80%) was obtained as a greenish-brown powder. m.p. 183–184 °C; ^1^H NMR (300 MHz, DMSO) δ 9.68 (d, *J* = 1.8 Hz, 1H), 9.60 (d, *J* = 1.9 Hz, 1H), 8.54 (d, *J* = 9.8 Hz, 1H), 7.76 (d, *J* = 7.9 Hz, 2H), 7.31–7.16 (m, 5H), 5.41 (t, *J* = 7.4 Hz, 2H), 4.17 (s, 3H), 3.33 (t, *J* = 7.3 Hz, 2H), 2.71 (s, 3H); ^13^C NMR (75 MHz, DMSO) δ 193.96, 166.85, 148.67, 145.78, 141.08, 136.49, 134.18, 129.16, 128.57, 127.10, 127.01, 124.40, 122.96, 99.38, 57.29, 34.71, 26.99; HRMS (ESI): *m*/*z* calcd. for C_20_H_20_NO_2_: 306.1494 [M-Br]^+^, found 306.1489. Anal. Calcd. for C_20_H_20_NO_2_Br: C, 62.19; H, 5.22; N, 3.63. Found: C, 62.07; H, 5.27; N, 3.59.

*3-Acetyl-7-methoxy-1-propylquinolin-1-ium bromide* (**2p**)

According to procedure B, from quinoline **6h** (150 mg, 0.745 mmol), propyl bromide (1.0 mL), dry acetonitrile (4.0 mL), 2 days. Quinolinium salt **2p** (170 mg, 70%) was obtained as a beige powder. m.p. 183–184 °C; ^1^H NMR (300 MHz, DMSO) δ 9.84 (d, *J* = 1.9 Hz, 1H), 9.72 (d, *J* = 1.8 Hz, 1H), 8.57 (d, *J* = 9.7 Hz, 1H), 7.79 (d, *J* = 7.8 Hz, 2H), 5.10 (t, *J* = 7.5 Hz, 2H), 4.17 (s, 3H), 2.80 (s, 3H), 2.07–1.94 (m, 2H), 0.99 (t, *J* = 7.3 Hz, 3H); ^13^C NMR (75 MHz, DMSO) δ 194.26, 166.83, 148.75, 145.74, 141.03, 134.29, 127.40, 124.51, 122.69, 99.49, 58.44, 57.40, 27.11, 22.29, 10.57; HRMS (ESI): *m*/*z* calcd. for C_15_H_18_NO_2_: 244.1338 [M-Br]^+^, found 244.1333. Anal. Calcd. for C_15_H_18_NO_2_Br: C, 55.57; H, 5.60; N, 4.32. Found: C, 55.67; H, 5.47; N, 4.25.

*General procedure C for the synthesis of compounds***6b**–**h**

Tin(II) chloride dihydrate was added to a solution of the corresponding 2-nitrobenzaldehyde derivative **3b**–**h** in EtOH. Thereafter, 4,4-dimethoxybutan-2-one **4** was introduced. The reaction mixture was heated to reflux for 4 h. After cooling to room temperature, the reaction mixture was concentrated and the residue was dissolved in EtOAc and hydrolyzed with saturated aqueous potassium sodium tartrate. The resulting emulsion was stirred overnight and the liquid was decanted. The organic phase was separated and the remaining aqueous layer was extracted with EtOAc. The combined organic layers were washed with brine, dried on MgSO_4_, and evaporated. The residue was purified by flash chromatography on silica gel using a mixture of petroleum ether/Et_2_O/EtOAc (ratio 5:3:2 to 4:3:3).

*3-Acetyl-7-trifluoromethylquinoline* (**6b**)

According to procedure C, 4-trifluoromethyl-2-nitrobenzaldehyde **3b** (1.0 g, 4.6 mmol), 4,4-dimethoxybutan-2-one **4** (1.51 g, 11.4 mmol), SnCl_2_·2H2O (4.12 g, 18.25 mmol) in ethanol (30 mL). Quinoline **6b** (0.425 g, 39%) was obtained as a beige powder. m.p. 102–103 °C; ^1^H NMR (300 MHz, CDCl_3_) δ 9.53 (d, *J* = 2.2 Hz, 1H), 8.77 (dd, *J* = 2.2, 0.8 Hz, 1H), 8.48 (dd, *J* = 1.8, 0.9 Hz, 1H), 8.10 (d, *J* = 8.6 Hz, 1H), 7.82 (dd, *J* = 8.6, 1.8 Hz, 1H), 2.79 (s, 3H); ^13^C NMR (75 MHz, CDCl_3_) δ 196.34, 150.43, 148.74, 136.92, 133.26 (q, *J* = 32.9 Hz), 130.55, 129.01, 128.34, 127.19 (q, *J* = 4.4 Hz), 125.40, 123.20 (q, *J* = 3.1 Hz), 26.86; ^19^F NMR (282 MHz, CDCl_3_) δ −62.94; HRMS (ESI): *m*/*z* calcd. for C_12_H_9_NOF_3_: 240.0636 [M + H]^+^, found 240.0634. Anal. Calcd. for C_12_H_8_NOF_3_: C, 60.26; H, 3.37; N, 5.86. Found: C, 60.16; H, 3.47; N, 5.77.

*3-Acetyl-7-dimethylaminoquinoline* (**6c**)

According to procedure C, 4-dimethylamino-2-nitrobenzaldehyde **3c** (1.0 g, 5.15 mmol), 4,4-dimethoxybutan-2-one **4** (1.70 g, 12.9 mmol), SnCl_2_·2H_2_O (4.65 g, 20.6 mmol) in ethanol (30 mL). Quinoline **6c** (0.480 g, 44%) was obtained as a beige powder. m.p. 150–151 °C; ^1^H NMR (300 MHz, CDCl_3_) δ 9.24 (d, *J* = 2.3 Hz, 1H), 8.48 (dd, *J* = 2.3, 0.8 Hz, 1H), 7.71 (d, *J* = 9.0 Hz, 1H), 7.22–7.07 (m, 2H), 3.14 (s, 6H), 2.65 (s, 3H); ^13^C NMR (75 MHz, CDCl_3_) δ 196.63, 152.94, 151.92, 150.27, 136.92, 130.48, 125.77, 118.97, 116.63, 106.22, 40.38, 26.62; HRMS (ESI): *m*/*z* calcd. for C_13_H_15_N_2_O: 215.1184 [M + H]^+^, found 215.1183. Anal. Calcd. for C_13_H_14_N_2_O: C, 72.87; H, 6.59; N, 13.07. Found: C, 72.99; H, 6.60; N, 13.23.

*3-Acetyl-7-bromoquinoline* (**6d**)

According to procedure C, 4-bromo-2-nitrobenzaldehyde **3d** (1.0 g, 4.35 mmol), 4,4-dimethoxybutan-2-one **4** (1.44 g, 10.9 mmol), SnCl_2_·2H_2_O (3.92 g, 17.4 mmol) in ethanol (30 mL). Quinoline **6d** (0.780 g, 72%) was obtained as a beige powder. m.p. 126–127 °C; ^1^H NMR (300 MHz, CDCl_3_) δ 9.43 (d, *J* = 2.2 Hz, 1H), 8.69 (dd, *J* = 2.3, 0.9 Hz, 1H), 8.36 (dd, *J* = 1.8, 0.9 Hz, 1H), 7.83 (dt, *J* = 8.7, 0.5 Hz, 1H), 7.73 (dd, *J* = 8.7, 1.9 Hz, 1H), 2.75 (s, 3H); ^13^C NMR (75 MHz, CDCl_3_) δ 196.44, 150.28, 150.24, 137.23, 132.01, 131.38, 130.54, 129.52, 126.69, 125.56, 26.95; HRMS (ESI): *m*/*z* calcd. for C_11_H_9_NO^79^Br: 249.9868 [M + H]^+^, found 249.9873. Anal. Calcd. For C_11_H_8_NOBr: C, 52.83; H, 3.22; N, 5.60. Found: C, 52.86; H, 3.32; N, 5.75.

*3-Acetyl-6-chloroquinoline* (**6e**)

According to procedure C, 5-chloro-2-nitrobenzaldehyde **3e** (3.0 g, 16.2 mmol), 4,4-dimethoxybutan-2-one **4** (5.34 g, 40.4 mmol), SnCl_2_·2H_2_O (14.59 g, 64.7 mmol) in ethanol (90 mL). Quinoline **6e** (2 g, 60%) was obtained as a beige powder. M.p. 146–147 °C; ^1^H NMR (300 MHz, CDCl_3_) δ 9.40 (d, *J* = 2.2 Hz, 1H), 8.61 (dd, *J* = 2.2, 0.8 Hz, 1H), 8.09 (dt, *J* = 9.0, 0.7 Hz, 1H), 7.92 (d, *J* = 2.3 Hz, 1H), 7.76 (dd, *J* = 9.0, 2.3 Hz, 1H), 2.74 (s, 3H);^13^C NMR (75 MHz, CDCl_3_) δ 196.56, 149.52, 148.27, 136.42, 133.57, 133.00, 131.19, 129.98, 127.92, 127.65, 27.04; HRMS (ESI): *m*/*z* calcd. For C_11_H_9_NO^35^Cl: 206.0373 [M + H]^+^, found 206.0368. Anal. Calcd. for C_11_H_8_NOCl: C, 64.25; H, 3.92; N, 6.81. Found: C, 64.55; H, 3.82; N, 6.90.

*3-Acetyl-6,7-dimethoxyquinoline* (**6f**)

According to procedure C, 6-nitroveratraldehyde **3f** (2.0 g, 9.5 mmol), 4,4-dimethoxybutan-2-one **4** (3.13 g, 23.7 mmol), SnCl_2_·2H_2_O (8.55 g, 37.9 mmol) in ethanol (60 mL). Quinoline **6f** (2.1 g, 96%) was obtained as a yellow powder. m.p. 218–219 °C; ^1^H NMR (300 MHz, CDCl_3_) δ 9.23 (d, *J* = 2.2 Hz, 1H), 8.56 (d, *J* = 2.2 Hz, 1H), 7.46 (s, 1H), 7.14 (s, 1H), 4.06 (s, 3H), 4.03 (s, 3H), 2.71 (s, 3H). ^13^C NMR (75 MHz, CDCl_3_) δ 196.90, 154.73, 150.67, 147.62, 147.52, 135.27, 128.20, 122.70, 108.05, 106.31, 56.48, 56.31, 26.90. HRMS (ESI): *m*/*z* calcd. for C_13_H_14_NO_3_: 232.0970 [M + H]^+^, found 232.0974. Anal. Calcd. for C_13_H_13_NO_3_: C, 67.52; H, 5.67; N, 6.06. Found: C, 67.56; H, 5.52; N, 6.00.

*7-Acetyl-*[1,3]*dioxolo[4,5-g]quinoline* (**6g**)

According to procedure C, 6-nitropiperonal **3g** (1.0 g, 5.1 mmol), 4,4-dimethoxybutan-2-one **4** (1.69 g, 12.8 mmol), SnCl_2_·2H_2_O (4.63 g, 20.5 mmol) in ethanol (30 mL). Quinoline **6g** (0.178 g, 16%) was obtained as a beige powder. m.p. 165–166 °C; ^1^H NMR (300 MHz, CDCl_3_) δ 9.22 (d, *J* = 2.2 Hz, 1H), 8.52 (dt, *J* = 2.2, 0.6 Hz, 1H), 7.43 (d, *J* = 0.6 Hz, 1H), 7.16 (s, 1H), 6.17 (s, 2H), 2.71 (s, 3H); ^13^C NMR (75 MHz, CDCl_3_) δ 196.82, 152.97, 148.89, 148.75, 147.47, 135.68, 128.12, 124.16, 105.94, 103.91, 102.38, 26.89; HRMS (ESI): *m*/*z* calcd. for C_12_H_10_NO_3_: 216.0661 [M + H]^+^, found 216.0653. Anal. Calcd. for C_12_H_9_NO_3_: C, 66.97; H, 4.22; N, 6.51. Found: C, 67.13; H, 4.42; N, 6.40.

*3-Acetyl-7-methoxyquinoline* (**6h**)

According to procedure C, 4-methoxy-2-nitrobenzaldehyde **3h** (1.0 g, 5.5 mmol), 4,4-dimethoxybutan-2-one **4** (1.82 g, 13.8 mmol), SnCl_2_·2H_2_O (4.98 g, 22.1 mmol) in ethanol (30 mL). Quinoline **6h** (0.71 g, 64%) was obtained as a beige powder. m.p. 129–130 °C; ^1^H NMR (300 MHz, CDCl_3_) δ 9.32 (d, *J* = 2.2 Hz, 1H), 8.58 (dd, *J* = 2.2, 0.8 Hz, 1H), 7.77 (d, *J* = 9.0 Hz, 1H), 7.41 (d, *J* = 2.6 Hz, 1H), 7.22 (dd, *J* = 9.0, 2.5 Hz, 1H), 3.95 (s, 3H), 2.68 (s, 3H); ^13^C NMR (75 MHz, CDCl_3_) δ 196.69, 162.91, 151.97, 149.80, 136.93, 130.51, 127.64, 122.00, 121.02, 107.48, 55.80, 26.79; HRMS (ESI): *m*/*z* calcd. for C_12_H_12_NO_2_: 202.0868 [M + H]^+^, found 202.0863. Anal. Calcd. for C_12_H_11_NO_2_: C, 71.63; H, 5.51; N, 6.96. Found: C, 71.76; H, 5.63; N, 7.07.

*Methyl 7-methoxyquinoline-3-carboxylate* (**6j**)

Methanol (0.9 mL, 22.26 mmol), TMSCl (1.18 g, 10.86 mmol), and quinoline **6i** (1.0 g, 5.43 mmol) were sequentially added to a dry flask under inert atmosphere at rt. The reaction mixture was heated at 50 °C for 4 h. After being cooled to rt, water (0.2 mL) was added to the mixture and followed by the addition of Na_2_CO_3_ (0.58 g) and CH_2_Cl_2_ (30 mL). The organic layer was dried on MgSO_4_ and concentrated under vacuum to afford 770 mg of ester **6j** (65%). ^1^H NMR (300 MHz, CDCl_3_) δ 8.94 (d, *J* = 2.1 Hz, 1H), 8.40 (dd, *J* = 2.1, 0.8 Hz, 1H), 7.75 (d, *J* = 9.0 Hz, 1H), 7.44 (d, *J* = 2.5 Hz, 1H), 7.30 (dd, *J* = 9.0, 2.5 Hz, 1H), 3.98 (s, 3H); ^13^C NMR (75 MHz, CDCl_3_) δ 163.41, 151.11, 150.51, 140.70, 129.44, 122.06, 121.67, 117.63, 107.80, 104.14, 55.97. Anal. Calcd. for C_12_H_11_NO_3_: C, 66.35; H, 5.10; N, 6.45. Found: C, 66.37; H, 5.05; N, 6.51.

*Ethyl 7-methoxyquinoline-3-carboxylate* (**6k**)

To a suspension of **5** (1.10 g, 5.41 mmol) in ethanol was added, dropwise, SOCl_2_ (2.76 mL, 38 mmol). The resulting mixture was stirred under reflux for 18 h. After concentration under reduced pressure, water (50 mL) was added and the crude mixture was neutralized to pH 7 with 20% solution Na_2_CO_3_. The aqueous solution was extracted with CH_2_Cl_2_ (3 × 50 mL), the combined organic layers were dried on MgSO_4_ and concentrated under vacuum to afford 940 mg (75%) of the desired product **6k**.^1^H NMR (300 MHz, CDCl_3_) δ 9.32 (d, *J* = 2.1 Hz, 1H), 8.67 (d, *J* = 2.1 Hz, 1H), 7.73 (d, *J* = 9.0 Hz, 1H), 7.40 (d, *J* = 2.4 Hz, 1H), 7.19 (dd, *J* = 9.0, 2.5 Hz, 1H), 4.41 (q, *J* = 14.1, 6.9 Hz, 2H), 3.93 (s, 3H), 1.41 (t, *J* = 7.2 Hz, 3H); ^13^C NMR (75 MHz, CDCl_3_) δ 165.57, 162.58, 151.84, 150.49, 138.03, 130.08, 121.98, 121.23, 120.71, 107.40, 61.25, 55.64, 14.35. Anal. Calcd. for C_13_H_13_NO_3_: C, 67.52; H, 5.67; N, 6.06. Found: C, 67.46; H, 5.72; N, 6.18.

*7-Methoxy-N,N-dimethylquinoline-3-carboxamide* (**6l**)

A 20 mL glass tube was charged with 7-methoxyquinoline-3-carboxylic acid **5** (0.5 g, 2.46 mmol), DMF (2 mL), T3P^®^ in DMF 50% (1.74 g, 2.71 mmol) and (CH_3_)_2_N·HCl (2 M in Et_2_O, 0.62 mL) at 20°C. The tube was sealed, heated to 130 °C and stirred for 18 h. The solution was carefully hydrolyzed at 10 °C with aqueous saturated Na_2_CO_3_ (5 mL) and extracted with ethyl acetate (4 × 20 mL). Combined organic phases were dried on MgSO_4_ and concentrated under reduced pressure. The crude product (light brown oil) was used without further purification. m.p. 110–111 °C; ^1^H NMR (300 MHz, CDCl_3_) δ 8.76 (d, *J* = 2.2 Hz, 1H), 8.03 (dd, *J* = 2.2, 0.8 Hz, 1H), 7.56 (d, *J* = 9.0 Hz, 1H), 7.30–7.24 (m, 1H), 7.07 (dd, *J* = 8.9, 2.5 Hz, 1H), 3.79 (s, 3H), 2.99 (s_br_, 3H), 2.93 (s_br_, 3H). ^13^C NMR (75 MHz, CDCl_3_) δ 169.10, 161.47, 149.77, 148.59, 134.73, 129.10, 126.70, 121.87, 120.46, 106.92, 55.37, 39.53, 35.37. HRMS (ESI): *m*/*z* calcd. for C_13_H_15_N_2_O_2_: 231.1134 [M + H]^+^, found 231.1132.

*N,7-Dimethoxy-N-methylquinoline-3-carboxamide* (**6m**)

To a suspension of 7-methoxyquinoline-3-carboxylic acid **5** (635 mg, 3.1 mmol) in dry DMF (4 mL), CDI (597 mg, 3.4 mmol) was added and the resulting mixture was stirred at 20 °C under a stream of argon for 1 h. Thereafter, DMF (2.0 mL), *N*,*O*-dimethylhydroxylamine hydrochloride (396 mg, 4.1 mmol) and triethylamine (0.44 mL, 3.1 mmol) were successively added and the mixture was stirred for 18 h. The brown suspension was diluted with ethyl acetate (50 mL) and washed with water. The aqueous layer was extracted twice with ethyl acetate and the organic layers were combined. The organic phase obtained was washed with a saturated solution of NaHCO_3_ (2 × 50 mL), water (2 × 50 mL), saturated solution of NH_4_Cl (2 × 50 mL), brine. The organic layer was dried on MgSO_4_ and concentrated under vacuum to afford 594 mg of **6m** as a pale orange oil (76%). ^1^H NMR (300 MHz, CDCl_3_) δ 9.18 (d, *J* = 2.1 Hz, 1H), 8.55 (d, *J* = 2.1 Hz, 1H), 7.79 (d, *J* = 8.9 Hz, 1H), 7.49 (d, *J* = 2.5 Hz, 1H), 7.26 (dd, *J* = 9.0, 2.5 Hz, 1H), 3.99 (s, 3H), 3.58 (s, 3H), 3.44 (s, 3H); ^13^C NMR (75 MHz, CDCl_3_) δ 167.63, 162.14, 150.63, 149.88, 136.94, 129.86, 124.73, 122.12, 120.62, 107.24, 61.32, 55.71, 33.39; HRMS (ESI): *m*/*z* calcd. for C_13_H_15_N_2_O_3_: 247.1083 [M + H]^+^, found 247.1082. Anal. Calcd. for C_13_H_14_N_2_O_3_: C, 63.40; H, 5.73; N, 11.38. Found: C, 63.61; H, 5.84; N, 11.48.

## 4. Conclusions

The search of new and readily accessible dual inhibitors of DYRK1A and CLK1, two kinases involved at different levels in the formation of amyloid plaques and neurofibrillary tangles in AD patients, remains a challenge in the development of new drug candidates in AD treatment. To address this issue, we designed herein a new class of dual inhibitors of DYRK1/CLK1 made up of a simple 1,4-dihydroquinoline structure, straightforwardly accessible in only three steps with good overall yields from commercial compounds. A preliminary SAR investigation highlighted that both an acetyl group at C-3 and a methoxy group at C-7 were required within the 1,4-dihydroquinoline scaffold to display the inhibition of *h*DYRK1A/*h*CLK1 kinases. Gratifyingly, compound **1p** proved to be the most potent, exhibiting three- to two-digit nanomolar IC_50_ values (153 nM and 47 nM toward *h*DYRK1A and *h*CLK1, respectively). Interestingly, dihydroquinoline **1p** exhibited antioxidant and radical scavenging properties (scavenging activity EC_50_ = 128 μM). This is a major asset over most existing inhibitors since oxidative stress is involved in the pathogenesis of AD. The ADME profile obtained for **1p** is highly encouraging (predicting inter alia good BBB permeability). Finally, molecular docking studies showed that *N*-methyl-3-acetyl-7-methoxy 1,4-dihydroquinoline **1h** reproduces the INDY binding mode in both kinases *h*DYRK1A and *h*CLK1. Overall, this work paves the way for further research efforts aiming at developing this novel class of dual-inhibitors of *h*DYRK1A and *h*CLK1 with in vitro antioxidant properties as promising drug candidates to treat AD.

## Data Availability

Not applicable.

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
