# Peer review of "Straightforward Access to a New Class of Dual DYRK1A/CLK1 Inhibitors Possessing a Simple Dihydroquinoline Core"

_molecules, 2022, doi:10.3390/molecules28010036_

Round 1

Reviewer 1 Report

In this manuscript entitled “Straightforward Access to a New Class of Dual DYRK1A/CLK1 Inhibitors Possessing a Simple Dihydroquinoline Core”. Vincent Levacher and co-workers describe a new class of dual inhibitors against AD.

The authors designed and synthesized a new type of dual inhibitor for DYRK1/CLK1 with 1,4-dihydroquinoline motif. The synthetic strategy is very straightforward, with only three steps required to complete the synthesis. The IC50s for inhibition of hDYRK1A/hCLK1 kinases are acceptable, given they are in nanomolar ranges. Interestingly, they have found compound 1p to possess antioxidant properties that might benefit the result of battling AD as oxidative stress is linked to the pathogenesis of AD. 

The SAR studies from this work indicated that the acetyl group at C3 position and the methoxy group at C7, and the examine-like nitrogen atom are required for the activity of the compound.

In this research, the authors have conducted in vitro experiments, docking studies, and in silico ADME evaluations. 

Overall, this work described promising results for a promising option against AD. I would recommend to accept it.

Author Response

Thank you for your recommendation to accept this research article.

Reviewer 2 Report

The article is interesting and leads to the identification of kinase inhibitors potentially useful for the treatment of SNC pathologies.

Nevertheless, in my opinion the study must be improved before its publication in Molecules.

In particular the ADME properties of the most active derivatives should be experimentally determined, at least their ability to cross the BBB.

Other minor points:

Page 2, line 50, the authors should explain what SR proteins are.

Page 2 line 54, it could be useful to add the chemical structures of some representative molecules

In the Scheme 1, which should be only Scheme, the substituents must be reported separately, not among the arrows.

In the Table 1, line 7, the R1 and R2 substituents must be reported in the same way used for the other compounds in the proper column.

While it is useless to report the well-known Lipinsky rule, the principle of the boiling egg is not clear and should be explained.

Round 2

Reviewer 2 Report

The authors performed the corrections suggested by the reviewers.

In my opinion the manuscript is acceptable in this form
